# *Lycium barbarum* (Wolfberry) Branches and Leaves Enhance the Growth Performance and Improve the Rumen Microbiota in Hu Sheep

**DOI:** 10.3390/ani14111610

**Published:** 2024-05-29

**Authors:** Pingping Duan, Halidai Rehemujiang, Lidong Zhang, Mulong Lu, Changchang Li, Lihong Hu, Youli Wang, Qiyu Diao, Guishan Xu

**Affiliations:** 1College of Animal Science and Technology, Tarim University, Alar 843300, China; 15770021628@163.com (P.D.); halidai@caas.cn (H.R.); 15899315771@163.com (L.Z.); a2774148614@163.com (M.L.); 18095365644@163.com (C.L.); hx18699756542@163.com (L.H.); 2College of Animal Science and Veterinary, Southwest Minzu University, Chengdu 610041, China; wangylwy@163.com; 3Institute of Feed Research, Key Laboratory of Feed Biotechnology of the Ministry of Agriculture and Rural Affairs, Chinese Academy of Agricultural Sciences, Beijing 100080, China; diaoqiyu@caas.cn; 4Key Laboratory of Tarim Animal Husbandry Science and Technology, Tarim University, Alar 843300, China

**Keywords:** *Lycium barbarum* branches and leaves, growth performance, slaughtering performance, rumen, microbiota

## Abstract

**Simple Summary:**

Simple Summary: *Lycium barbarum* (Wolfberry) extract is predominantly utilized in livestock and poultry applications. However, the impact of unprocessed branches and leaves of *Lycium barbarum* as a roughage source on ruminant growth performance, slaughter metrics, and rumen micro-organisms remains not well understood. This investigation examines the effects of incorporating *Lycium barbarum* branches and leaves into the diet of Hu sheep, specifically assessing growth performance, slaughter results, and meat quality through feeding trials and slaughter tests. Additionally, this study explored the impact on rumen microflora by utilizing 16S rDNA sequencing technology. Findings indicated that diet inclusion of *Lycium barbarum* branches and leaves could enhance nutrient absorption, which in turn improves growth performance and certain aspects of meat quality.

**Abstract:**

The *Lycium barbarum* branches and leaves (LBL) are known to contain a range of active substances that have positive effects on animal immunity and antioxidation. This study aimed to examine how LBL impacts the growth and slaughter performance as well as rumen fermentation and microbiota in Hu sheep. A total of 50 male Hu sheep of indigenous origin, aged 3 months, were randomly divided into 5 groups of 10 sheep each. The groups were given different levels of LBL supplementation (0%, 3%, 6%, 9%, and 12%) to evaluate growth performance and nutrient apparent digestibility. Rumen fluid samples were collected for analysis of the fermentation parameters and rumen chyme was examined to study the rumen microbiota. The slaughter performance, meat quality, and organ index were evaluated at the conclusion of the experiment. The results showed that the final body weight and average daily gain of the LBL1 group were significantly higher than those of the CON group, LBL3 group, and LBL4 group (*p* < 0.05). The average dry matter intake of the LBL4 group was significantly lower than that of other experimental groups (*p* < 0.05). The apparent digestibility of CP in the LBL1 and LBL2 groups was higher than that in other experimental groups (*p* < 0.05). At the same time, the eye muscle area and grade-rule (GR) value of Hu sheep in the LBL1 group significantly increased and the quality of Hu sheep meat improved (*p* < 0.05). There was no significant difference in organ weight and organ index between the experimental groups (*p* > 0.05). The pH of the rumen fluid in the LBL1 group was significantly lower than that in the CON group (*p* < 0.05). There was no significant difference in the NH_3_-N content between the experimental groups (*p* > 0.05). The propionate and valerate in the rumen fluid of Hu sheep in the LBL2 group were significantly higher than those in other experimental groups (*p* < 0.05). In addition, this had no significant effect on the structure and abundance of the rumen microbiota (*p* > 0.05). LBL is a promising functional feed. Adding an appropriate amount of LBL to the diet can improve the feed efficiency, growth performance, and meat quality of Hu sheep but has no adverse effects on the rumen. In this experiment, the appropriate supplemental level of LBL in the diet was 3%.

## 1. Introduction

*Lycium barbarum* (Wolfberry) is a medicinal plant from the Solanaceae family, widely distributed in arid and semi-arid regions [1]. In 2020, China’s wolfberry production is expected to have reached 441,200 tons, accounting for over 95% of the world’s total. The plant is renowned for its abundant nutrients and bioactive compounds [2,3]. The *Lycium barbarum* plant has been used in China for over 2000 years as an herbal medicine and dietary supplement [4]. In traditional Chinese medicine, and as a functional food, *Lycium barbarum* is widely used, with the fruit, leaves, and root bark all being used for this purpose [5,6]. *Lycium barbarum* is a plant whose fruit, branches, and leaves are rich in nutrients and active ingredients such as polysaccharides [7,8], carotenoids, betaine, tannins, phenols, flavonoids [9] as well as a variety of amino acids and trace elements [10]. These bioactive factors enhance immunity through humoral regulation, promote bone proliferation and development [11], and improve the body’s oxidative capacity by regulating enzyme activity [12]. Furthermore, they suppress the expression of proinflammatory cytokines, thereby reducing inflammatory responses [13]. The polysaccharides from *Lycium barbarum* leaves profoundly affect the gut microbiota [14].

The growth process of wolfberries involves constant pruning and shaping. Although annual harvests of wolfberry branches exceed 200,000 tons, the majority of them are either incinerated or wasted. Only a small number are used for research on cuttings and seedlings as well as cultivation substrates. This leads to a waste of resources and environmental pollution. It has been shown that *Lycium barbarum* leaves are rich in nutrients, including protein, minerals, nicotinic acid, polysaccharides, and flavonoids [15,16]. The leaves of *Lycium barbarum* are the main source of bioactive substances, with certain nutrients exceeding the contents found in the fruits [17]. Therefore, this suggests that the branches and leaves of *Lycium barbarum* (LBL) can serve as livestock forage. Research has demonstrated that supplementing *Lycium barbarum* into the diets of livestock and poultry boosts their growth performance and immune response [18], enhances meat quality [19], and regulates the structure of gastrointestinal flora [20].

However, the current utilization of LBL as animal feed is low, leading to significant waste in some areas. The sheep industry has developed rapidly over the past few years and there has been an exploration of unconventional feed resources. Currently, there is limited research on the use of LBL in sheep production, with previous studies primarily focusing on the utilization of *Lycium barbarum* fruits and extracts. The aim of this research is to explore how varying levels of LBL in the diets of Hu sheep affect their growth and slaughter performance as well as rumen fermentation and rumen microbiota. This study will also analyze the nutritional properties of LBL, offering a theoretical foundation for its use in sheep farming.

## 2. Materials and Methods

### 2.1. Ethics Committee Approval

The study was carried out in accordance with the procedures sanctioned for this research, which have been approved by the Science and Technology Ethics Committee of Tarim University (ethics number 2024049). These procedures adhere to the principles and regulations for ethical protection in human and animal biological science and technology in China.

### 2.2. Animals Study Design and Diets

The trial was scheduled to take place between March 2023 and June 2023. Research was carried out at the Alar-based Tarim University Animal Science Experiment Station in southern Xinjiang, China. An experimental design that was completely randomized was employed. A total of 50 Hu sheep, all healthy males aged 3 months with good body condition and an average weight of 21.39 ± 1.82 kg, were chosen. These sheep were randomly split into five groups according to their weight with ten sheep in each group. There were five diets with different proportions of LBL that were provided for the five groups as follows: the control group (CON, 0% LBL), test group 1 (LBL1, 3% LBL), test group 2 (LBL2, 6% LBL), test group 3 (LBL3, 9% LBL), and test group 4 (LBL4, 12% LBL). In this experiment, the branches and leaves of two-year-old *Lycium barbarum* plants were collected in April 2022. The ratio of branches to leaves was 3:1 and the dry matter content was 93.74%. Under the dry material basis, the crude protein content was 16.75%, crude fat 6.1%, neutral detergent fiber 42.97%, acid detergent fiber 27.74%, crude ash 9.71%, calcium 1.46%, and phosphorus 0.22%. The diets were formulated in accordance with the guidelines provided by the China Feed Research Institute as outlined in the Nutrient Requirements of Meat-Type Sheep and Goat (NY/T816-2021). Table 1 provides a comprehensive breakdown of the composition and nutrient levels. The sheep were placed in individual metabolism cages measuring 1.4 m × 0.7 m × 0.6 m. They were given food twice daily at 10:00 AM and 6:00 PM in addition to having unrestricted access to fresh water and a salt block. Prior to the commencement of the study, the animals were acclimatized to an experimental diet over a period of 10 days. The feeding trial spanned 80 days before sample collection. Ivermectin was administered to all animals at a dosage of 0.2 mg/kg of body weight to eradicate parasites.

#### 2.2.1. Determination Method of Nutrient Content

An analysis was conducted on samples of diet, ingredients, and feces for dry matter (method 930.15), CP (method 990.03), EE (method 920.39), Ca (method 978.02), and P (method 946.06) using the AOAC procedures [21]. NDF and ADF content were determined following Van Soest’s method [22]. ME was calculated based on the measured nutritional value [23]. The ME was calculated using the following formula:ME = 0.046 + 0.820 × (17.211 − 0.135 × NDF).

#### 2.2.2. Growth Performance and Apparent Digestibility

During the experimental period, the daily feed amount provided to each group of sheep was recorded to determine the average daily feed intake (ADFI). Furthermore, the sheep were weighed before feeding at intervals of 0, 15, 30, 45, 60, and 80 days to calculate their average daily gain (ADG). The ratio of feed to gain (F:G) was subsequently calculated by utilizing both ADG and ADFI.

Five sheep with similar body weights (BW) were selected for each treatment group. Digestibility trials were conducted from day 40 to 49, lasting for 10 days, which included a 5-day adaptation period and a 5-day period for collecting whole feces. Feces were collected daily for 5 consecutive days during the digestibility test. Ten percent of the total feces was collected from each sheep and divided into two equal portions. One portion was mixed with 10% H_2_SO_4_ to assess the digestibility of CP. The remaining portion was utilized to assess the digestibility of various nutrients. The feces from each sheep were gathered over a span of 5 days, then combined, dried at a temperature of 65 °C, and subsequently ground using a 1 mm sieve for further examination. Feed and fecal samples were collected on a daily basis, mixed together, dried at 65 °C for a duration of 72 h, and finally sieved using a 1 mm mesh.

#### 2.2.3. Rumen Fermentation Parameters and Micro-Organisms

Immediately upon slaughter, the ruminal fluid was gathered. After the rumen contents were filtered through sterile 4-layer gauze, 2 50 mL portions of rumen fluid were extracted. The pH of the rumen fluid in a 50 mL centrifuge tube was tested using a portable pH meter (PHS-3C, Shanghai, China) and the measurement was repeated 3 times. The rumen fluid in the other 50 mL centrifuge tube was centrifuged for 15 min at 4000 r/min to collect the supernatant, which was then divided into 3 15 mL centrifuge tubes. Subsequently, 8 mL of rumen supernatant was taken and 2 mL of a freshly prepared 25% metaphosphoric acid solution was added. The sample was frozen at −20 °C for the determination of ammonia nitrogen (NH_3_-N). Then, 4 mL of rumen supernatant was collected, 1 mL of the previously mentioned 25% metaphosphoric acid solution was added, and the sample was frozen at −20 °C for the examination of volatile fatty acids (VFAs). VFAs, including acetate, propionate, butyrate, and valerate, were analyzed by HPLC (Agilent 7890A, Santa Clara, CA, USA).

An extra 2 mL of rumen contents was obtained and moved into freezing tubes. The samples were then immediately frozen in liquid nitrogen and kept in a freezer at −80 °C. This procedure was performed to enable subsequent DNA extraction and high-throughput sequencing analysis.

#### 2.2.4. Extraction of DNA and Sequencing of 16S rDNA

The TGuide S96 Magnetic Bead Method Soil/Fecal Genomic DNA Extraction Kit (Tiangen, Beijing, China) was used to extract DNA from the rumen specimens for 16S rDNA sequencing analysis. The concentration of the extracted nucleic acids was determined using an enzyme marker (GeneCompany Limited, Hong Kong, China, model Synergy HTX), and their integrity was assessed through agarose electrophoresis at a concentration of 1.8% (Beijing Bomei Fuxin Technology Co., Ltd., Beijing, China).

The highly variable V3-V4 regions of bacterial 16S rDNA were amplified by PCR using universal bacterial primers 338F (5′- ACTCCTACGGGGAGGCAGCA-3′) and 806R (5′- GGACTACHVGGGGTWTCTAAT-3′) after extracting total DNA from the samples. The final products were cleaned, quantified, and mixed to form a sequencing collection. After passing quality assessments, the collection was analyzed on the Illumina NovaSeq 6000 instrument (San Diego, CA, USA). Raw data files from various high-throughput sequencers, including the Illumina NovaSeq, were processed to generate sequenced reads. These reads include both the actual sequences and quality metrics. The initial reads were refined with the Trimmomatic v0.33 tool. To further refine the data, Cutadapt 1.9.1 was utilized to eliminate primer sequences and produce polished reads.

#### 2.2.5. Slaughter Performance and Meat Quality

Upon completion of the feeding trial, five sheep were chosen at random from each treatment group and fasted for a period of 16 h prior to measuring their live weight prior to slaughter (LWBS). Carcass weight refers to the weight of the sheep’s body without the hide, head, carpal joints of the front legs, the lower part of the hock joint in the back legs, and internal organs (excluding kidneys and kidney fat) post-slaughter. The dressing percentage was determined using the subsequent equation:Dressing percentage (%) = carcass weight/live weight before slaughter × 100.

The research assessed the slaughter characteristics by quantifying the eye muscle area, backfat thickness, and Grade-rule (GR) value. The eye muscle area represents the cross-sectional size of the longissimus dorsi muscle situated between the second rib and the penultimate rib of the carcass. The computation formula for the eye muscle area is given by the following: 0.7 × eye muscle height × eye muscle width. Backfat thickness refers to the layer of fat positioned above the center of the eye muscle between the 12th and 13th ribs. The GR value serves as a measure of the carcass’s fat content, defined as the total tissue thickness between the 12th and 13th ribs, taken 11 cm from the carcass’s dorsal midline.

Furthermore, the heart, liver, spleen, lungs, kidneys, and pancreas weights were measured for every sheep. Additionally, the weights of the rumen, reticulum, omasum, abomasum, and small intestine were recorded post content removal. These measurements were utilized to determine the organ index, which was computed using the subsequent equation:Organ index (%) = organ weight/live weight before slaughter × 100

A total of 200 g of sheep meat from the longissimus dorsi muscle between the 11th and 13th ribs was collected post-slaughter for quality evaluation. The samples underwent analysis for water loss, pH levels, cooking loss, shear force, and color. Meat samples weighing around 5 g were packed with filter papers, pressed with a 35 kg square iron for 5 min, and then weighed post pressing. The calculation for water loss percentage is as follows: (initial meat weight–final meat weight)/initial meat weight × 100. pH levels were determined using a portable pH meter (Testo 205; Testo AG, Titisee-Neustadt, Germany) after allowing the meat to stand for 45 min and 24 h. Prior to measuring, the pH meter was calibrated with standard pH buffers (pH 4.00, 6.86, and 9.18). For cooking loss assessment, 30 g of meat was cooked in boiling water for 30 min then weighed after cooling at room temperature for 20 min in the shade. Cooking loss was calculated as the weight after cooking/weight before cooking × 100. Meat samples were heated to 80 °C in a water bath and removed at 70 °C core temperature. Per NY/T 1236-2006, samples were cut vertically into 1 cm^3^ blocks along the myofiber direction for shear force measurement using a meat tenderizer (C-LM3B; Tenovo International Co., Ltd, Beijing, China). Color (L*, a*, b*) readings were taken after the meat bloomed for 15 min using a portable colorimeter (CR-10; Minolta, Tokyo, Japan).

### 2.3. Statistical Analysis

The data set was organized using Excel 2010 and a single-factor analysis of variance (ANOVA) was conducted with SPSS 23.0 for statistical analysis. The results from the tests were expressed as the mean and the standard error of the mean (SEM), with *p* < 0.05, indicating a significant difference, and 0.05 < *p* < 0.10, indicating a trend. 

The DATA2 [24] method in QIIME2 2020.6 [25] was primarily used for sequence data analysis. This involved denoising, bipartite sequence splicing, removal of chimeric sequences, and clustering at a 97% similarity threshold to generate operational taxonomic units (OTUs). The taxonomy annotation of the OTUs was conducted by classifying representative organisms using a Bayesian classifier based on the SILVA database (version 138). Alpha diversity analyzes species diversity and complexity using ACE, Chao1, Simpson, and Shannon indices. Beta diversity analyses were conducted using principal coordinate analysis (PCoA) and nonmetric multidimensional scaling (NMDS) to evaluate distinctions between groups. Tax4Fun was used to predict the KEGG functions based on pairs of 16S rDNA data.

## 3. Results

### 3.1. Growth Performance and Apparent Digestibility

As shown in Table 2, there were no significant differences in the initial weight among the different groups (*p* > 0.05). However, the final body weight and ADG were significantly higher in the LBL1 group (*p* < 0.05). Lambs in the LBL4 group had a significantly lower dry matter intake (DMI) compared to the other experimental groups (*p* < 0.05). Additionally, dietary LBL supplementation did not affect the F:G ratio (*p* > 0.05). The CP digestibility was high in the LBL1 and LBL2 groups compared to the other experimental groups (*p* < 0.05). Nevertheless, the addition of LBL did not result in any notable impact on the apparent digestion of DM, NDF, and ADF within all test subjects throughout the duration of the study (*p* > 0.05).

### 3.2. Rumen Fermentation Parameters

As shown in Table 3, the pH value of the LBL1 group was significantly lower than that of the control group (*p* < 0.05). All other test groups showed a decreasing trend in the concentration of NH_3_-N in the sheep rumen fluid compared to the control group (*p* = 0.079). The LBL2 group had significantly higher levels of propionate and valerate in the rumen fluid compared to the other experimental groups (*p* < 0.05). However, no significant differences were found in the levels of total volatile fatty acids (TVFA), acetate, butyrate, and the acetate to propionate ratio (A:P) among the experimental groups.

### 3.3. Effects of Abundance, Diversity, and Composition of Rumen Bacteria

The sequencing coverage for each sample group exceeded 99%, ensuring that the collected samples accurately reflected changes in the rumen microbiota. Sequences were clustered at a 97% similarity, resulting in 18,945 OTUs across the 24 samples (Figure 1). A total of 601 OTUs were shared between samples from different groups. The LBL4 group had a higher number of OTUs compared to the other treatment groups. Alpha diversity analyzes species diversity and complexity, including ACE, Chao1, Simpson, and Shannon indices (Appendix A). The ACE, Chao1, and Shannon indices showed an upward trend compared to the control group, suggesting that the addition of LBL in the diet the ration increased the richness and diversity of the rumen microbial community (*p* = 0.079, Figure 2A; *p* = 0.081, Figure 2B; *p* = 0.077, Figure 2C). However, the Simpson index did not show a significant difference among the groups (Figure 2D).

A study on beta-diversity was conducted to explore differences in the rumen microbial populations across the various cohorts. The PCoA analysis of the Bray-Curtis distance matrix (Figure 3A) revealed that the representative points of rumen micro-organisms in different treatment groups were dispersed across separate quadrants within the coordinate plane. This suggests that the addition of LBL did not significantly affect the species and abundance of micro-organisms in the rumen. NMDS analyses differ from PCoA analyses in that they use points in space to represent different samples, reflecting the differences between samples as distances between points in space. The NMDS analysis (Figure 3B) based on the Bray-Curtis distance matrix indicated that the green dots of the LBL2 group were dispersed and distant from each other, whereas the purple dots of the LBL4 group were closely clustered. It is suggested that the rumen bacterial flora of the LBL4 group had a greater evolutionary similarity compared to the LBL2 group.

The taxonomic annotation of feature sequences was performed using a plain Bayesian classifier resulting in the identification of 42 bacterial phyla and 1334 bacterial genera from 24 samples of gastrointestinal tract contents from Hu sheep. The *Firmicutes* and *Bacteroidetes* were the predominant phyla detected in all samples, collectively accounting for an average of 80% of the bacterial community (Figure 4A). The bacterial phyla mainly include *Spirochaetota*, *Fibrobacterota*, *Patescibacteria*, *Verrucomicrobia*, *Proteobacteria*, *Actinobacteria*, and *Cyanobacteria*. The proportion of *Spirochaetota* seemed to increase in the LBL-supplemented groups (*p* < 0.05). No significant differences were observed in the proportion of other bacteria between the LBL-treated sheep and the control group (*p* > 0.05) (Appendix A). At the genus level, the most prevalent genera in each treatment group were *Prevotella*, *Rikenellaceae_RC9_gut_group*, *Succiniclasticum*, *Saccharofermentans*, *Treponema*, and *Fibrobacter* (Figure 4B). However, there were no significant differences in the relative abundance of dominant genera between the groups (*p* > 0.05).

Utilizing the Tax4Fun technique enabled the prediction of KEGG functions for prokaryotic microbial communities based on 16S high-throughput sequencing data’s species classification. The research did not detect any significant functional differences among the experimental groups (*p* > 0.05) (Figure 5A). Nevertheless, more than 70% of the enrichment focused on metabolic functions, specifically carbohydrate metabolism and membrane transport (at the KEGG pathway level 1). Figure 5B illustrates the identification of 46 functional classes (at the KEGG pathway class 2 level) associated with rumen bacteria. These classes primarily revolved around metabolism, including carbohydrate, amino acid, energy, cofactor, microbial, lipid, glycan biosynthesis and metabolism, terpenoids, and polyketides metabolism. Furthermore, a few connections were made with drug resistance. Concurrently, the examination of KEGG pathways at a tertiary level showed predicted functional pathways linked to rumen bacteria, such as antibiotic biosynthesis, secondary metabolite biosynthesis, and additional metabolic pathways (Figure 5C).

### 3.4. Slaughter Performance, Meat Quality, and Organ Development

As indicated in Table 4, there was no observed effect of dietary LBL supplementation on the LWBS, carcass weight, slaughter rate, and back fat thickness (*p* > 0.05). The LBL1 group showed a significant increase in the eye muscle area of Hu sheep (*p* < 0.05), whereas no significant difference was noted between the control and LBL3 groups. The GR value of Hu sheep was significantly reduced by 12% with supplementation compared to the control lambs (*p* < 0.05).

The meat in the LBL1 group had a significantly higher water loss rate compared to the other groups (*p* < 0.05). There was no significant effect of dietary LBL on the shear force, cooking loss, and pH value at 45 min (*p* > 0.05). The pH value decreased over 24 h in the 3% LBL supplementation groups compared to the control group (*p* < 0.05). The L* color value of the LBL1 group showed a decreasing trend compared to the control and LBL4 groups (*p* < 0.05). The a* and b* color values were not affected by LBL supplementation (*p* > 0.05).

The weights of internal organs were measured after the slaughter. As shown in Table 5, supplementation with LBL did not significantly affect the weights of the heart, liver, spleen, lung, kidney, pancreas, rumen, reticulum, omasum, abomasum, and small intestine (*p* > 0.05).

## 4. Discussion

*Lycium barbarum* is a traditional Chinese herb that contains bioactive substances in its fruit, branches, leaves, and root bark. These substances have medicinal value and promote antioxidant, antifatigue, immunomodulatory, neuroprotective, hypoglycemic, antiosteoporotic, and antitumor effects [26]. The key indicators for growth performance are average daily feed intake (ADI) and feed conversion ratio (FCR) [27]. A statistically significant correlation was found between FCR and growth performance [28]. Menchetti [29] found that the addition of 1% goji berries to rabbit diets led to a significant increase in final body weight and feed efficiency, a decrease in mortality, and an improvement in reproductive performance. Hao et al. [30] discovered that supplementing feed with 1% wolfberry and astragalus extracts significantly improved the growth performance of fattening pigs. Furthermore, it regulated the unsaturated fatty acid content of pork, thereby improving its nutritional value. The results of this experiment showed that the addition of LBL to the diet could significantly increase the final body weight and ADG of Hu sheep and the 3–6% addition group had the best effect. In addition, compared to other experimental groups, the F:G ratio of the LBL1 group was the lowest, consistent with findings from previous studies [31]. This suggests that including LBL promotes the growth of Hu sheep, possibly due to the richness of LBL in polysaccharides, carotenoids, polyphenols, and flavonoids. These compounds enhance the fermentation of micro-organisms in the rumen [32] and improve the organism’s ability to absorb nutrients, promoting the animal’s growth performance.

The digestibility of nutrients indicates an animal’s ability to digest and utilize the nutrients in its diet to a certain extent [33]. Differences in digestibility are correlated with dry matter and nutrient intake [34]. Jiang et al. [35] conducted nylon bag tests to evaluate the rumen digestibility of seven herbs and their potential to meet the nutritional requirements of ruminants. Zheng et al. [31] found that feeding 10 g *Lycium barbarum* polysaccharide daily could significantly increase the apparent digestibility of nutrients and promote growth performance in calves. The inclusion of 3% LBL in the diet significantly enhanced the apparent digestibility of CP in the diet of Hu sheep. However, it did not significantly affect the digestibility of other nutrients, which is consistent with previous studies. The analysis was conducted to determine the reason for this finding. It was discovered that *Lycium barbarum* polysaccharide (LBP), an active ingredient found in LBL, can regulate intestinal flora, leading to an increase in the relative abundance of beneficial flora in the intestinal micro-organisms [36]. This promotes the digestion and absorption of nutrients [37]. As a result, it indirectly promotes animal growth and development by increasing the digestibility of CP.

The rumen, which is the primary organ for digesting feed in ruminants, houses a large number of complex micro-organisms that provide nutrients and energy to the host through mutual collaboration [38]. The key indicators of microbial fermentation in the rumen of ruminant animals are pH, NH_3_-N, and VFAs [39]. The growth and multiplication of rumen micro-organisms can be directly influenced by pH. To ensure optimal protein synthesis in rumen micro-organisms, it is recommended to maintain the pH of rumen between 5.9 and 7.2 [40]. NH_3_-N levels are closely linked to bacterial protein levels and serve as a key nitrogen source for protein synthesis in rumen micro-organisms [41]. Rumen fermentation produces VFAs, which are a primary source of energy for ruminants. Acetic, propionic, butyric, and valeric acids play significant roles in rumen microbial metabolism [42]. Efficient lambs can improve feed utilization without compromising rumen volatile fatty acids [43]. In this study, the inclusion of LBL in the diet resulted in decreased pH and NH_3_-N levels in the rumen fluid of Hu sheep. This finding is consistent with the results of Paya et al. [43], who reported a decrease in NH3-N levels in sheep rumen following the addition of inulin to the diet. Simultaneously, the inclusion of 3% LBL in the diet significantly increased the concentration of propionate and valerate in the rumen of Hu sheep. This outcome aligns with the findings of Zhang et al. [32], who demonstrated that incorporating LBL into the diet can elevate the concentration of propionate in the rumen of Hu sheep. In summary, the inclusion of 3% LBL in the ratio enhanced rumen fermentation in Hu sheep. This improvement can be linked to the presence of active compounds in LBL that stimulate the growth of bacteria responsible for producing short-chain fatty acids. This leads to an increase in VFAs concentration and a decrease in pH, creating an acidic environment that fosters the growth of beneficial micro-organisms in the intestines. As a result, the intestinal microbial community and function are altered and the pathogenic bacterial flora is reduced [44].

The rumen is a diverse ecosystem that plays a crucial role in the digestion and breakdown of nutrients consumed by ruminants [45]. It is home to a variety of micro-organisms, such as bacteria, archaea, anaerobic fungi, protozoa, and phages, that collaborate in the decomposition of fermented feed to supply essential nutrients and energy to the host [46]. Research indicates that the rumen microbiome is affected by host genetics and has a significant impact on host health and growth [47,48]. Furthermore, the composition of the diet is a major factor that influences the rumen microbial profiles [49]. The stability of the rumen microbiota is essential for the growth and health of ruminants. The microbiota develops from birth and influences the nutritional balance, digestion, and metabolism of ruminants [50]. Studies consistently show that *Firmicutes* and *Bacteroidota* are the dominant phyla in the gastrointestinal tract of ruminants and some monogastric animals. In both cows and sheep, *Firmicutes* and *Bacteroidetes* are the most prevalent phyla, as discovered by Wang et al. [51] and Mani et al. [52]. *Bacteroidetes* aid in the breakdown of complex carbohydrates, improving nutrient digestion and absorption [53]. *Bacteroidetes* are capable of producing various vitamins that contribute to different rumen functions, such as amino acid and lipid catabolism, thereby enhancing the host’s lipolytic metabolism [54,55]. *Firmicutes*, on the other hand, are associated with obesity and promote the breakdown of cellulose, xylan, and pectin in the gastrointestinal tract of animals [56]. They play a crucial role in the metabolism of rumen substances. The rumen of Hu sheep was found to be dominated by the *Firmicutes* and *Bacteroidota* phyla, which accounted for over 80% of the relative abundance [57]. In this experiment, it has been shown that the main phyla present in all samples were *Firmicutes*, *Bacteroides*, *Spirochaetota*, *Fibrobacterota*, and *Patescibacteria*. Research has indicated that, despite its small genome, the *Spirochaetota* displays a broad spectrum of potential functional variety, adjusting to the breakdown of intricate polysaccharides within plant cell walls [58], in addition to synthesizing B-complex vitamins and breaking down proteins in the rumen [59]. Our investigation revealed that the proportionate abundance of *Spirochaetota* in the LBL group surpassed that in the CON group. These findings suggest that LBL has the ability to enhance the multiplication and expansion of *Spirochaetota* in the rumen, hasten the breakdown of plant polysaccharides and proteins in the diet, increase the production of propionate, supply more energy for the organism, and stimulate animal growth.

The dominant genera of rumen bacteria composition at the genus level are consistent with previous studies [60]. *Prevotella* and *Rikenellaceae_RC9_gut_group* had a the highest relative abundance in the rumen at the genus level, followed by *Succiniclasticum*, *Saccharofermentans*, *Treponema*, and *Fibrobacter*. The rumen bacterial community structure is similar to that of the goat community [61], indicating a high concentration of fiber degradation. Although micro-organisms are not abundant, they can aid in digestion nutrient absorption, and provide energy to the organism. *Prevotella* can digest and use nutrients from plant sources such as hemicellulose and starch [62]. It also participates in protein degradation and amino acid metabolism [63]. *Rikenellaceae_RC9_gut_group* shows a positive correlation with the growth performance of host animals. This has the ability to degrade plant-derived cellulose and hemicellulose, converting them into short-chain fatty acids. These fatty acids can be absorbed by the host and utilized for energy, particularly for the production of propionate fatty acids [64]. The presence of *Rikenellaceae_RC9_gut_group* indicates a healthy gastrointestinal system [65], with a relative abundance above 5% in this experiment. The study results suggest that the inclusion of LBL in the diet did not significantly affect the relative abundance of dominant bacterial genera in the rumen of Hu sheep.

The health and metabolism of the host are closely related to the development of rumen microbes. The functional pathways involved in rumen microbes in this experiment were primarily related to metabolism, including carbohydrate metabolism, amino acid metabolism, and energy metabolism. This is in line with previous studies conducted on Tibetan sheep [66] and Hu sheep [67]. The addition of LBL did not have a significant impact on rumen microbiota-related functions and pathways compared to the control. Upon analysis of KEGG metabolic pathways at the tertiary level, it was discovered that antibiotics accounted for up to 4% synthesized. No antibiotics were synthesized in the KEGG metabolic pathway during the third level of analysis in the study of the effect of tea saponin on the rumen function of Qinchuan beef cattle [68]. The reason may be attributed to the active substances in LBL and there are few studies on this aspect that need to be explored in the future.

Slaughter performance is a vital indicator for measuring the production performance of meat animals. The feed of fattened lambs influences their growth, carcass, and meat characteristics [69]. It has been shown that the quality of meat can be enhanced by adding herbal supplements to an animal’s diet [70,71]. Castrica et al. [72] found that including 3% goji berry in the diet of female rabbits did not significantly affect body weight at slaughter or carcass weight. However, it improved the organoleptic properties of the rabbit meat. Menchetti et al. [73] discovered that adding at least 3% goji berries to the diet of female rabbits increased the phenolic content and antioxidant properties of the rabbit meat. Ju et al. [74] found that adding 0.3% *Lycium barbarum* polysaccharide to the diet could improve the growth performance, feed efficiency, and meat quality of lambs. In these experimental conditions, there were no significant differences in preslaughter live weight, carcass weight, slaughter rate, and backfat thickness among all groups of Hu sheep. However, including 3–6% LBL in the diet significantly improved the water loss rate and meat color of the meat. Possible reasons for differences in the results of previous studies may include variations in sheep breeds, differences in the form and concentration of the active substance in *Lycium barbarum* by-products added to the diet, and the necessity for further investigation into the specific mechanism of the observed effect.

The development of an animal’s internal organs and gastrointestinal tract are closely related. The study found that adding LBL to the daily ration did not significantly affect the weights and indices of internal organs and the gastrointestinal tract of the Hu sheep in all groups. This indicates that adding LBL to the daily ration did not have any adverse effects on the organ development of the Hu sheep. This is consistent with the results of Ju et al. [74]’s addition of LBP to the diet and found no significant difference in organ index between the treatment groups. However, there were also contrasting findings. For instance, the addition of 4 g/kg LBP to broiler diets promoted the development of immune organs and improved growth performance [75]. The variability in the results may be due to differences in the test animals’ organisms, as confirmed by the nonsignificant differences in the final body weights of the Hu sheep among the groups.

## 5. Conclusions

In conclusion, LBL shows potential as a functional feed. The inclusion of LBL in the diet has the potential to enhance growth performance, feed efficiency, and meat quality in Hu sheep. Importantly, no adverse effects were noted on rumen microbiota across all treatment groups. We suggest that supplemental level of LBL in the Hu sheep diet was 3 %.

## Figures and Tables

**Figure 1 animals-14-01610-f001:**
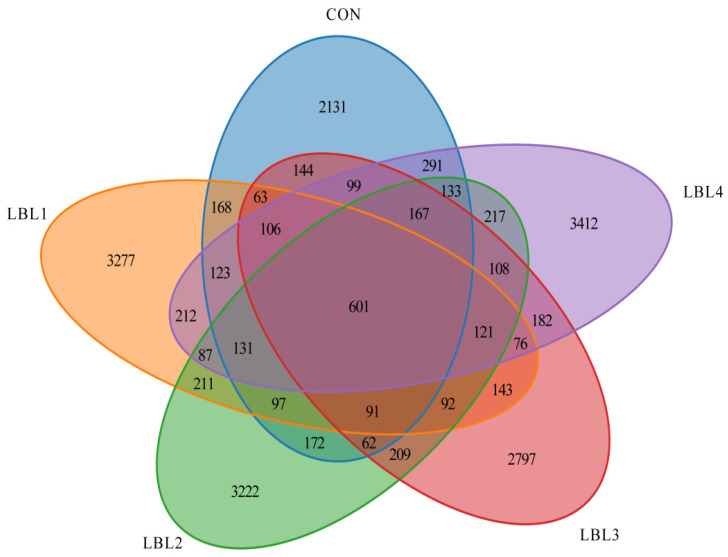
Venn diagram of ruminal contents flora.

**Figure 2 animals-14-01610-f002:**
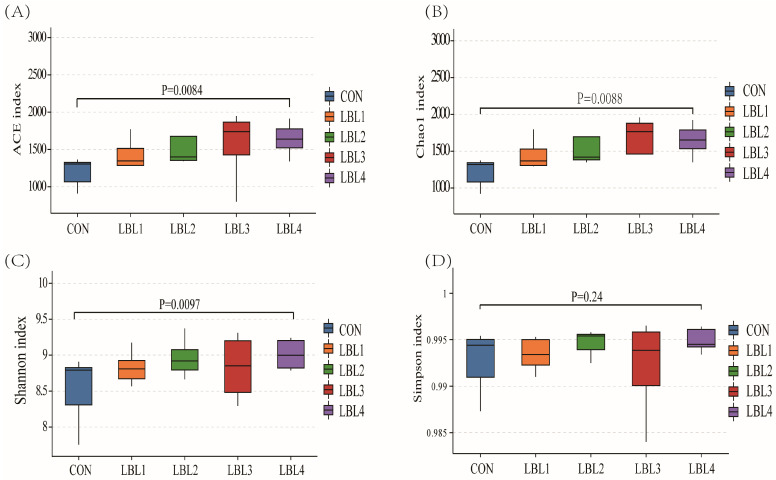
Alpha diversity analysis of rumen flora. (**A**) ACE index of species richness; (**B**) Chao1 index of species richness; (**C**) Shannon index of species diversity; (**D**) Simpson index of species diversity.

**Figure 3 animals-14-01610-f003:**
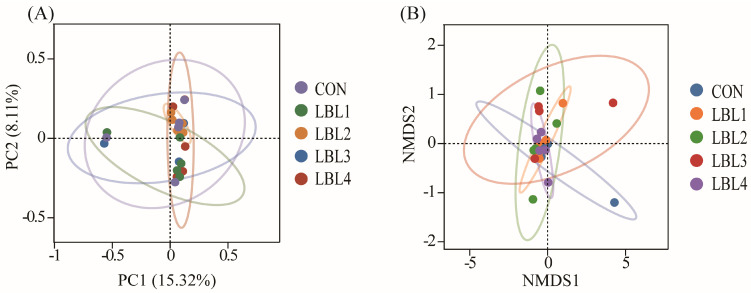
Beta diversity analysis of rumen flora. (**A**) PCoA principal axis analysis; (**B**) NMDS non-metric multidimensional scaling analysis.

**Figure 4 animals-14-01610-f004:**
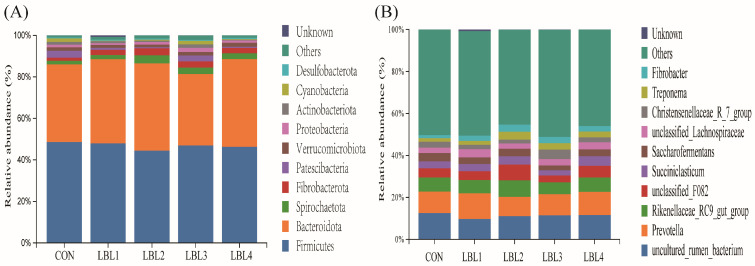
Distribution of bacterial taxa averaged under phyla (**A**) and genera (**B**) levels across the different treatment groups (as a percentage of the total sequence).

**Figure 5 animals-14-01610-f005:**
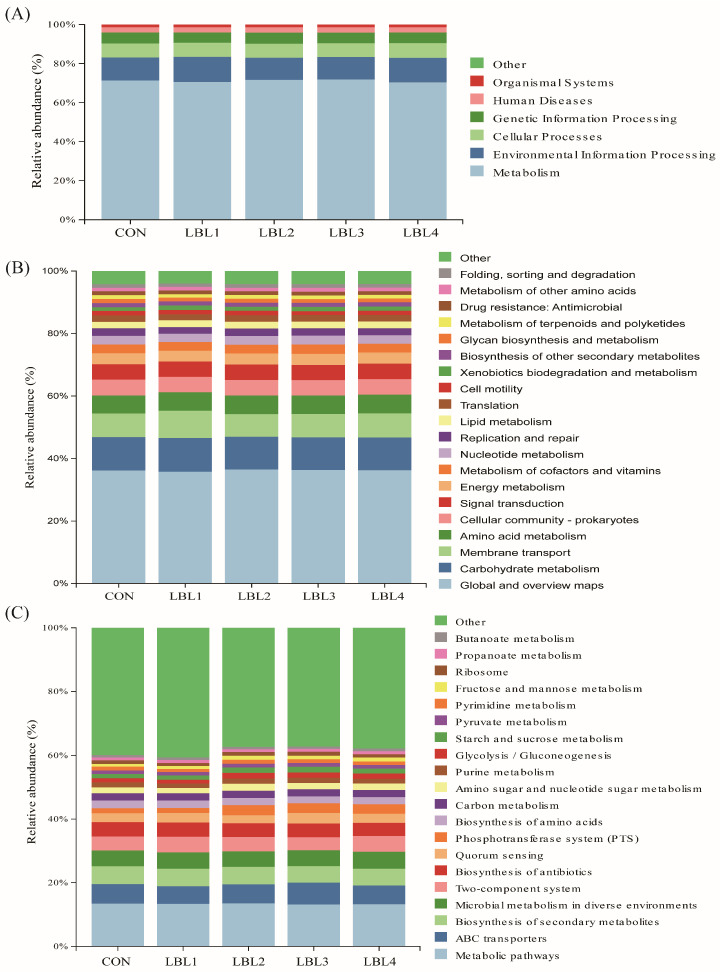
(**A**) Prediction of relative abundance of ruminal bacteria metabolic functions by Level 1 Kyoto Encyclopedia of Genes and Genomes (KEGG); (**B**) Prediction of relative abundance of ruminal bacteria metabolic functions by Level 2 KEGG; (**C**) Prediction of relative abundance of ruminal bacteria metabolic functions by Level 3 KEGG.

**Table 1 animals-14-01610-t001:** Composition and nutrition levels of the basal diet (dry matter basis, %).

Items	Group				
	CON	LBL1	LBL2	LBL3	LBL4
Ingredients (%)					
LBL	0.00	3.00	6.00	9.00	12.00
Alfalfa meal	9.00	6.00	3.00	1.50	1.00
Whole corn silage	5.50	5.20	5.00	4.30	3.50
Corn stalk	45.50	45.80	46.00	45.20	43.50
Corn	12.00	12.00	12.00	12.00	12.00
Wheat bran	7.00	7.00	7.00	7.00	7.00
Soybean meal	4.00	4.00	4.00	4.00	4.00
cottonseed meal	12.70	12.70	12.70	12.70	12.70
CaHPO_4_	1.30	1.30	1.30	1.30	1.30
NaCl	1.00	1.00	1.00	1.00	1.00
Premix ^1^	2.00	2.00	2.00	2.00	2.00
Nutrient components ^2^					
ME (MJ/kg)	9.75	9.88	9.73	9.53	9.42
CP (%)	13.72	13.69	13.68	13.74	13.87
EE (%)	9.70	10.07	9.32	9.69	9.86
NDF (%)	39.75	38.65	40.03	41.82	42.84
ADF (%)	24.14	23.81	24.19	24.17	24.02
Ca (%)	0.98	0.98	0.98	0.99	1.01
P (%)	0.66	0.66	0.66	0.66	0.66

^1^ The premix per kilogram diet contained 28,000 IU vitamin A, 4000 IU vitamin D3, 16 IU vitamin E, 0.2 mg vitamin K3, 16 mg niacin, 16 mg copper, 120 mg zinc, 0.3 mg selenium, 36 mg iron, 0.6 mg iodine, and 5.6 mg manganese. ^2^ Metabolizable energy was calculated and other nutrient components were measured.

**Table 2 animals-14-01610-t002:** Effects of LBL on the growth performance and nutrient apparent digestibility of Hu sheep (*n* = 10).

Items ^1^	CON	LBL1	LBL2	LBL3	LBL4	SEM	*p*-Value
Growth performance							
Initial BW, kg	20.32	20.53	20.48	20.26	20.38	0.258	0.782
Final weight, kg	33.90 ^b^	35.46 ^a^	34.30 ^ab^	33.83 ^b^	32.40 ^c^	0.116	0.032
ADG, g	169.75 ^b^	186.56 ^a^	172.75 ^ab^	169.56 ^b^	150.25 ^c^	3.531	0.022
ADFI, kg	1.43 ^a^	1.43 ^ab^	1.45 ^a^	1.47 ^a^	1.35 ^b^	0.013	0.047
F:G	8.46	7.69	8.43	8.69	8.98	0.200	0.068
Digestibility							
DM, %	70.45	72.31	72.16	71.17	71.38	0.813	0.553
CP, %	58.68 ^c^	65.62 ^a^	67.81 ^a^	60.16 ^b^	62.27 ^b^	0.617	0.036
NDF, %	49.37	51.08	51.57	49.68	52.49	1.671	0.621
ADF, %	38.24	42.69	42.07	39.83	38.24	1.441	0.416

^1^ BW, body weight; ADG, average daily gain; ADFI, average daily feed intake; F:G, ADFI/ADG. ^a,b,c^ Different superscripts indicate significant differences within a row (*p* < 0.05). SEM is the pooled standard error between five groups; the *p*-value indicates significance.

**Table 3 animals-14-01610-t003:** Effects of LBL on rumen fermentation parameters in Hu sheep (*n* = 5).

Items	CON	LBL1	LBL2	LBL3	LBL4	SEM	*p*-Value
pH	7.05 ^a^	6.72 ^b^	6.99 ^ab^	7.02 ^a^	6.84 ^ab^	0.044	0.037
NH_3_-N	15.29	14.33	14.25	14.20	14.51	0.175	0.079
TVFA	87.87	94.85	94.45	90.13	91.62	1.168	0.095
Acetate	56.31	60.31	60.14	57.35	59.27	1.122	0.346
Propionate	19.64 ^c^	22.21 ^a^	21.40 ^ab^	20.16 ^bc^	19.85 ^c^	0.278	0.004
Butyrate	10.93	11.26	11.89	11.62	11.53	0.141	0.054
Valerate	0.99 ^bc^	1.07 ^a^	1.02 ^b^	1.00 ^bc^	0.97 ^c^	0.009	0.000
A:P	2.88	2.71	2.82	2.84	3.01	0.066	0.229

^a,b,c^ Different superscripts indicate significant differences within a row (*p* < 0.05). SEM is the pooled standard error between five groups; the *p*-value indicates significance.

**Table 4 animals-14-01610-t004:** Effects of LBL on slaughter performance and meat quality in Hu sheep (*n* = 5).

Items	CON	LBL1	LBL2	LBL3	LBL4	SEM	*p*-Value
Slaughter performance							
LWBS, kg	35.84	36.98	36.55	36.18	33.87	0.452	0.053
Carcass weight, kg	15.62	16.46	16.34	16.24	15.04	0.220	0.093
Dressing percentage, %	43.70	44.55	44.73	44.98	44.46	0.584	0.571
Eye muscle area, cm^2^	19.84 ^bc^	21.80 ^a^	20.46 ^b^	19.94 ^bc^	18.94 ^c^	0.237	0.000
GR value, mm	5.04 ^b^	5.93 ^a^	5.91 ^a^	4.66 ^c^	4.13 ^d^	0.120	0.002
Backfat thickness, mm	4.56	4.66	4.60	4.40	4.33	0.082	0.710
Meat quality							
Water loss rate, %	19.97 ^bc^	21.12 ^a^	20.82 ^ab^	19.81 ^c^	19.67 ^c^	0.172	0.009
pH 0 h	6.25	6.30	6.35	6.31	6.24	0.038	0.438
pH 24 h	5.65 ^ab^	5.58 ^b^	5.66 ^ab^	5.69 ^ab^	5.79 ^a^	0.023	0.044
Cooking loss, %	26.08	25.97	27.51	27.30	27.42	0.390	0.287
Shear force, N	60.21	65.49	64.22	53.31	54.73	2.047	0.091
Shear force, kgf	6.16	6.70	6.54	5.45	5.60	0.209	0.088
L*	32.60 ^a^	28.53 ^c^	30.27 ^b^	31.13 ^ab^	32.87 ^a^	0.401	0.000
a*	14.13	16.00	15.67	16.47	14.53	0.412	0.117
b*	19.33	20.40	20.20	19.60	19.47	0.160	0.052

^a,b,c,d^ Different superscripts indicate significant differences within a row (*p* < 0.05). SEM is the pooled standard error between five groups; the *p*-value indicates significance.

**Table 5 animals-14-01610-t005:** Effects of LBL on organ index in Hu sheep (*n* = 5).

Items	CON	LBL1	LBL2	LBL3	LBL4	SEM	*p*-Value
Organ weight, g							
Heart	136.62	147.918	143.536	140.302	140.61	1.342	0.080
Liver	549.756	562.076	570.672	547.426	543.014	14.391	0.979
Spleen	42.744	48.108	45.636	45.434	45.058	1.021	0.628
Lung	525.87	564.324	558.034	528.176	536.776	9.665	0.655
Kidney	105.758	114.798	107.99	101.65	105.87	3.340	0.825
Pancreas	34.264	27.718	25.344	25.928	26.672	1.209	0.114
Rumen	664.524	665.408	675.166	669.442	658.42	10.426	0.993
Reticulum	96.664	103.298	106.848	94.834	100.618	2.178	0.430
Omasum	111.174	102.772	113.452	105.114	102.242	3.960	0.881
Abomasum	128.75	134.268	130.07	133.846	133.592	4.242	0.993
Small intestine	566.112	591.612	585.248	579.598	522.388	14.957	0.637
Organ index, %							
Heart	0.388	0.416	0.394	0.39	0.414	0.005	0.144
Liver	1.554	1.576	1.564	1.518	1.604	0.036	0.971
Spleen	0.12	0.134	0.126	0.124	0.132	0.003	0.644
Lung	1.49	1.582	1.538	1.466	1.58	0.028	0.625
Kidney	0.298	0.322	0.296	0.28	0.31	0.009	0.656
Pancreas	0.096	0.078	0.07	0.07	0.078	0.004	0.113
Rumen	1.882	1.878	1.856	1.856	1.934	0.030	0.988
Reticulum	0.272	0.292	0.292	0.262	0.294	0.007	0.434
Omasum	0.318	0.288	0.314	0.29	0.3	0.011	0.878
Abomasum	0.362	0.376	0.356	0.374	0.392	0.010	0.845
Small intestine	1.596	1.654	1.608	1.606	1.538	0.034	0.900

## Data Availability

The 16S rRNA data of rumen contents samples are available at the National Center for Biotechnology Information (NCBI) Sequence Read Archive under the accession number PRJNA1064231.

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
