# Peer review of "Lycium barbarum* (Wolfberry) Branches and Leaves Enhance the Growth Performance and Improve the Rumen Microbiota in Hu Sheep"

_animals, 2024, doi:10.3390/ani14111610_

Round 1

Reviewer 1 Report

Comments and Suggestions for Authors

Comments on the Quality of English Language

Author Response

Dear Editors and Reviewers:
Thank you for your comments concerning our manuscript entitled "Lycium barbarum branches and leaves enhance the growth performance and improve the rumen microbiota in Hu sheep”(ID: animals-2982614).Those comments are all valuable and very helpful for revising and improving our paper, as welll as the important guiding significance to our researches. We havestudied comments carefully and have made correction which we hope meet with approval. Revised portion are marked in red in the paper. Here are two other modifications that need to be explained to you : 1.We added two authors. 2.We modified the structure of the entire article by placing rumen fermentation parameters and microorganisms after growth performance.In addition, the main revisions of the paper and the responses to the reviewers ' comments are as flowing:

Response to Reviewer 1 Comments

Comments and response are as follows:

Point 1: Introduction: at line 62 please explain what is "yellow alcohol" .

Response 1:We changed "yellow alcohol substances" to "flavonoids". (L 63).

Methods:

Point 2: please check the text for tense. It would be helpful to describe the stage of maturity of the LBL forage and the proportions of leaves and branches in the material used. You mention "GR value" at line 131 but this is not defined until it is explained in the footnote to Table 3 and similarly,  "organ index" is mentioned in  line 135, but not defined until Table 4. Presumably the rumen liquor was collected, mixed and then centrifuged; please give more information about how this was done. Describe the meat quality measurements in the Methods, not in the Results.

Response 2: We have checked the tenses in the methods and they have all been in the past tense. We have considered your comments carefully and agree with your assessment. The stage of maturity of the LBL forage and the proportions of leaves and branches in the material used have been added  and labelled in red. (L111-115)

Point 3: At line 121 you say "The excrement of each sheep was collected"; please explain this as excrement seems to be different to faeces which is mentioned in the line 123.

Response 3: You are right. Thank you for your careful checks. We are sorry for our carelessness. It is important to note that “excreta” and “faeces” are not the same. During the digestion test, we collected all the faeces of each test sheep for one day, not the excreta. This was an error in our description in the text, which has now been changed to “faeces” and highlighted in red. (L146-150)

Point 4: You mention "GR value" at line 131 but this is not defined until it is explained in the footnote to Table 3 and similarly,  "organ index" is mentioned in  line 135, but not defined until Table 4. Presumably the rumen liquor was collected, mixed and then centrifuged; please give more information about how this was done. Describe the meat quality measurements in the Methods, not in the Results.

Response 4: We agree with the your suggestion and have rewritten this section accordingly. We have added specific explanations of the indicators of slaughter performance and meat quality and their determination, as well as a detailed description of the procedure for the treatment of rumen liquor. Detailed measurement information has been added to the method from page 3, line 136 to page 4, line 190.

Results:

Point 5: LBL may have some type of toxic or anti-nutritional properties because the 12% level gave significantly poorer results than the control diet in several cases. Figs 1A and 1B are diff to read; are these figures necessary or can they be deleted?  Their contents are adequately summed up in the text where you say  "The dilution  curve  in  Figure  1A  indicates  that  sequencing  coverage  has  reached saturation after 20,000 reads. The species accumulation plot in Figure 1B shows that as the sample size increases, both the number of species in the environment and the number of shared species reach saturation. These are comments on the efficacy of the methods that you have used, rather than results relevant to the use of LBL. Similarly, I suggest that Figure 2 is deleted. The rumen microbial data is interesting from an ecological point of view but no conclusions are drawn from that which would indicate why different levels of LBL have different effects; can this material be removed?

Response 5: Figures 1A and 1B proved the effectiveness of the methods used in this test, showed that further analysis can be carried out, and are not linked to the test results. We have removed Figures 1A and 1B at your suggestion, as well as the related descriptions. However, we have not deleted Figure 2 after much deliberation. Our team is convinced that the Wayne plots are also part of the microbiological data and to some extent reflect the differences between the different treatment groups.

Point 6: Discussion: please confine the discussion to animals which have been fed LBL. Responses to other herbs are not relevant. At line 367 you refer to LBP -what is it, and what evidence is there that it regulated gut microflora in this or any other experiment?

Response 6: Yes, we have deleted the relevant description according to your opinion, and the discussion will be more focused on the Lycium barbarum plants. LBP refers to Lycium barbarum polysaccharide, which has been added in the full name of LBP. We have also added references to LBP's role in gut flora to the article. (L398)

Point 7: Conclusions: you say that "In conclusion  the addition of 3% LBL had a positive effect on Hu sheep improving   growth    performance,   slaughter    performance,   meat    quality,   rumen    fermentation parameters,  without  impacting  the   diversity  and  abundance  of  rumen  flora,  and  resulting  in considerable economic benefits (not measured). Further investigation is required to determine the pecif cmechanism of this effect." I think this statement exaggerates the effect of LBL. Growth and slaughter performances were improved in only a very few aspects, LBL improved meat quality only in relation to water holding capacity, and the only fermentation parameter affected was the proportion of propionic acid (although there was a trend to increase the total VFA concentration). You did not measure economic performance and so the claim that this was improved can not be justified. I would suggest that it is important to standardise the proportions of leaves and branches in the supplement and also the size of the branch particles, and also the stage of maturity of the LBL plant. All of these will affect the nutrient and other constituent contents, and the possible effects of indigestible and difficult-to-chew material in the diet. Finally, many of the results that you have reported are, although statistically significant, quite small and may not have any great practical effect.

Response 7: This test did not directly measure the economic benefits; rather, it inferred them through the ratio of material to weight. Upon review, we found this statement to be inaccurate. Consequently, the conclusion has been modified, removing the phrase 'considerable economic benefits'. Additionally, we have included information about the mature stage, the proportion of branches and leaves, and the nutritional components of the collected wolfberry branches and leaves in the material method.

Reviewer 2 Report

Comments and Suggestions for Authors

Review Animals Lycium barbarum

General.

Please explain why you chose 5 levels of Lycium in the diet but did not use a polynomial analysis to test for the dose response to the level of Lycium inclusion. The trial design is inconsistent with the statistical model used and detracts from the value of the results. From the data it appears that the lowest inclusion level of Lycium provided the biggest response for most variables. Thereafter there was either no further response or a reduction in the size of the response relative to the control. The data has to be reanalysed using polynomial analysis to test for linear, quadratic or quartic response to Lycium inclusion level.  The Materials and methods are incomplete. Methods should be included in the text not included as footnotes to the tables. There is no mention of how the animals were housed. Were they group housed according to treatment of individually penned. This affects the replication for statistical analysis and needs to be included in the methods. From Line 110, it appears that all sheep for a particular inclusion level were housed together. This being the case there is no replication for animal.

Throughout the discussion you cite numerous examples where specific bioactive compounds have been identified in Lycium and animal responses to these bioactives have been demonstrated. In your study, these bioactives were not analysed. Therefore you cannot ascribe a response to a particular bioactive. Discussion on this is warranted and perhaps you can make reference to data from the literature where these bioactives have been quantified in Lycium leaves and stems.

The interesting conclusion I came to was that any response to Lycium is seen at low levels of inclusion in the diet. I would have thought this was a very encouraging result as it means benefits are achieved with just low levels of the supplement. You fail to make any comment on this.

Specific commnets

Line 106. Methods for nutrient analysis need to be included in the text.

Line 107. A reference for the ME calculation needs to be given.

Line 109. Need to provide details on housing, specifically noting if sheep were individually housed of housed in groups.

Line 129. Replace “fur” with “hide”

Line 131. You need to define GR and give details in the text, not as a footnote in the tables.

Line 135. Define organ index in the text.

Line 138. Define methods for meat quality analysis.

Line 197. Define LWBS in text. What is slaughter rate? Is this dressing percentage? Please either explain or standardise nomenclature.

Line 237 to 239. You discuss a decreasing trend compared to control. Does this mean that the P value is a linear response to a polynomial analysis? I cannot see this. The trend is quadratic with an initial increase in values to the 3% inclusion rate of Lycium followed by a decline in values. Again this problem can be resolved by choosing the appropriate statistical analysis.

Figure 1B. The variables are not identified. Also the text is too small to read.

Line 336 to 349. Here and elsewhere, you begin each section of the discussion with a collection of data from the literature followed by a repeat of your results with little actual discussion. Typically, one would begin with a discussion of your own results followed by relevant citations from the literature that either support or refute your findings. This would be a better way of structuring the discussion and you should consider changing your style here and throughout the discussion.

Line 338 to 339. I see no need to mention residual feed intake as you did not measure it.

Line 354 to 356. You need a reference to support this statement.

Line 366. The sentence beginning on this line is out of context and has no meaning.

Lines 337 to 374. The same phrase is repeated three times.

Lines 431 to 506. I am not an expert on the rumen microbiota so cannot comment.

Author Response

Dear Editors and Reviewers:
Thank you for your comments concerning our manuscript entitled "Lycium barbarum branches and leaves enhance the growth performance and improve the rumen microbiota in Hu sheep”(ID: animals-2982614).Those comments are all valuable and very helpful for revising and improving our paper, as welll as the important guiding significance to our researches. We havestudied comments carefully and have made correction which we hope meet with approval. Revised portion are marked in red in the paper. Here are two other modifications that need to be explained to you : 1.We added two authors. 2.We modified the structure of the entire article by placing rumen fermentation parameters and microorganisms after growth performance.In addition, the main revisions of the paper and the responses to the reviewers ' comments are as flowing:

Response to Reviewer 2 Comments

Comments and response are as follows:

General.

Please explain why you chose 5 levels of Lycium in the diet but did not use a polynomial analysis to test for the dose response to the level of Lycium inclusion. The trial design is inconsistent with the statistical model used and detracts from the value of the results. From the data it appears that the lowest inclusion level of Lycium provided the biggest response for most variables. Thereafter there was either no further response or a reduction in the size of the response relative to the control. The data has to be reanalysed using polynomial analysis to test for linear, quadratic or quartic response to Lycium inclusion level.  The Materials and methods are incomplete. Methods should be included in the text not included as footnotes to the tables. There is no mention of how the animals were housed. Were they group housed according to treatment of individually penned. This affects the replication for statistical analysis and needs to be included in the methods. From Line 110, it appears that all sheep for a particular inclusion level were housed together. This being the case there is no replication for animal.

Throughout the discussion you cite numerous examples where specific bioactive compounds have been identified in Lycium and animal responses to these bioactives have been demonstrated. In your study, these bioactives were not analysed. Therefore you cannot ascribe a response to a particular bioactive. Discussion on this is warranted and perhaps you can make reference to data from the literature where these bioactives have been quantified in Lycium leaves and stems.

The interesting conclusion I came to was that any response to Lycium is seen at low levels of inclusion in the diet. I would have thought this was a very encouraging result as it means benefits are achieved with just low levels of the supplement. You fail to make any comment on this.

Specific commnets

Point 1: Line 106. Methods for nutrient analysis need to be included in the text.

Response 1:Thank you for your comments. We have added the methods for nutrient analysis to the materials and methods section of the manuscript. (line127-132).

Point 2: Line 107. A reference for the ME calculation needs to be given.

Response 2: Thank you for your comments,we have added reference for ME calculation. (line 132-134)

Point 3: Line 109. Need to provide details on housing, specifically noting if sheep were individually housed of housed in groups.

Response 3: Thank you for your comments. We provided detail on housing, our sheep were individually adaptation. (Line 114-115)

Point 4: Line 129. Replace “fur” with “hide”

Response 4: Thank you for your comments. We replaced “fur” with “hide”.(line 190)

Point 5: Line 131. You need to define GR and give details in the text, not as a footnote in the tables.

Response 5:Thank you for your comments. We defined GR and gave details in the text. (line 200-201)

Point 6: Line 135. Define organ index in the text.

Response 6: Thank you for your comments. We defined organ index in the text. (line 207).

Point 7: Line 138. Define methods for meat quality analysis.

Response 7: Thank you for your comments. We defined methods for meat quality analysis in the text. (line 211-226).

Point 8:Line 197. Define LWBS in text. What is slaughter rate? Is this dressing percentage? Please either explain or standardise nomenclature.

Response 8 : Thank you for your comments. We defined LWBS in the text, and explained slaughter rate et al . (line 188-194).

Point 9: Line 237 to 239. You discuss a decreasing trend compared to control. Does this mean that the P value is a linear response to a polynomial analysis? I cannot see this. The trend is quadratic with an initial increase in values to the 3% inclusion rate of Lycium followed by a decline in values. Again this problem can be resolved by choosing the appropriate statistical analysis.

Response 9 : We used SPSS software for one-way analysis of variance for all test data. After analyzing the NH3-N concentration data, P value = 0.079, indicating that the difference between different treatment groups was not significant, but after careful observation of the data, it was found that there was a downward trend. Therefore, we finally discussed that the experimental group with wolfberry branches and leaves had a downward trend compared with the control group.

Point 10: Figure 1B. The variables are not identified. Also the text is too small to read.

Response 10 : Figures 1A and 1B proved the effectiveness of the methods used in this test, showed that further analysis can be carried out, and are not linked to the test results. We have removed Figures 1A and 1B at your suggestion, as well as the related descriptions.

Point 11: Line 336 to 349. Here and elsewhere, you begin each section of the discussion with a collection of data from the literature followed by a repeat of your results with little actual discussion. Typically, one would begin with a discussion of your own results followed by relevant citations from the literature that either support or refute your findings. This would be a better way of structuring the discussion and you should consider changing your style here and throughout the discussion.

Response 11:Thank you for your comments.We found that this part of the discussion is not well written, so according to your comments and other reviewers will re-write the discussion part.

Point 12: Line 338 to 339. I see no need to mention residual feed intake as you did not measure it.

Response 12::Thank you for your comments. We have deleted the residual feed intake.

Point 13:Line 354 to 356. You need a reference to support this statement.

Response 13:We have added references. (line 383).

Point 14: Line 366. The sentence beginning on this line is out of context and has no meaning.

Response 14:Thank you for your comments.We have deleted this part and rewritten it.

Point 15: Lines 337 to 374. The same phrase is repeated three times.

Response 15:Thank you for your comments.We have deleted this part.

Point 16:Lines 431 to 506. I am not an expert on the rumen microbiota so cannot comment.

Response 16:Thank you for your comments.We have revised the whole article according to the opinions of you and other reviewers.

Reviewer 3 Report

Comments and Suggestions for Authors

Title: Lycium barbarum branches and leaves enhance the growth peformance and improve the rumen microbiota in Hu sheep

Objective: Effects of Lycium barbarum branches and leaves on slaughter performance, growth performance and apparent digestibility of nutritent, meat quality and sequencing of Hu sheep.

Limitations: Check the various details mentioned in the comments and please also check why there is no explanation as to why the treatment with the lowest level of Lycium barbarum branches and leaves has an effect and the other concentrations show a lower effect on the studied variables.

Comments

Simple Summary: No comments

Abstract:

Delete the word “Background”. What is the meaning of GR value?

What is “positive effect on the growth, development? organ development” referring to?

Keywords: What is “slaughtering performance”? Does it refer to carcass quality?

Introduction: No comments

Materials and Methods

Lines 121-124: Is the process done twice?

Line 131: What does “GR value” mean?

Line 142: Was metaphosphoric acid added for the measuement of volatile fatty acids?

Results

Lines 211-226 and 234: The description of each variable obtained can be presented in the Materials and Methods section.

Table 5: The variable for the acetate/propionate ratio could be added.

Discussion

Line 351: The following statement “the F:G  ratio was significantly reduced” should be changed to “tends to” because the p-value was 0.068.

Please add the reference(s) for the following paragraph: “suggests that including LBL promotes the growth of Hu sheep, possibly due to the richness of LBL in polysaccharides, carotenoids, polyphenols, and flavonoids. These compounds enhance the fermentation of microorganisms in the rumen and improve the organism's ability to absorb nutrients, promoting the animal's growth performance.” 

Line 366: Cite previous studies.

Line 370: What does LBP mean? Please add the full name.

Lines 367-373: The following sentence is repeated three times. Please correct and add its reference(s). “It was discovered that LBP, an active ingredient found in LBL, can regulate intestinal flora, leading to an increase in the relative abundance of beneficial flora in the intestinal microorganisms.”

Lines 374-375: Add the corresponding citations.

Lines 342, 380, 383: The citations are made using the authors’ first names when it should be their last names. Please check and change all the corresponding citations in the whole document.

Line 392: In the following statement, “differences in the form”, does the word “form” refer to the type of active compound?

Lines 399-403: The arguments about LBP, even when they could be valid, are risky assumptions since there is no mention in the document on which concentrations of this compound were used in the treatment groups, neither theoretically nor practically.

Lines 422-424: Even when propionate and valerate were increased during the experiment, it doesn’t mean that the total VFA concentration was increased as argued with the following statement “This result is consistent with the findings of Zhang et al. [52], who demonstrated that adding LBL to the diet could increase the total concentration of VFA in the rumen of Hu sheep”.

Lines 427-430: The argument about the effect on the intestinal microorganisms has no basis or reference that supports it.

Lines 504-506: It is not the clear what the following sentence is supposed to expose “This suggests that an ingredient in the diets used in this study has the ability to promote the synthesis of antibiotics, although the specific mechanism of this effect is not yet known”.

Conclusions

What is meant with “considerable economic benefits”? What variable(s) does it refer to? Please consider to rewrite the Conclusions with a higher level of detail and specify the limitations of the study.

References

Please check and homogenize the format: should the journal name be abbreviated? Use the same uppercase and lowercase format for it. Also check which part should be italicized and write the authors’ names starting with their last names, instead of their first names

Author Response

Dear Editors and Reviewers:
Thank you for your comments concerning our manuscript entitled "Lycium barbarum branches and leaves enhance the growth performance and improve the rumen microbiota in Hu sheep”(ID: animals-2982614).Those comments are all valuable and very helpful for revising and improving our paper, as welll as the important guiding significance to our researches. We havestudied comments carefully and have made correction which we hope meet with approval. Revised portion are marked in red in the paper. Here are two other modifications that need to be explained to you : 1.We added two authors. 2.We modified the structure of the entire article by placing rumen fermentation parameters and microorganisms after growth performance.In addition, the main revisions of the paper and the responses to the reviewers ' comments are as flowing:

Response to Reviewer 3 Comments

Title: Lycium barbarum branches and leaves enhance the growth peformance and improve the rumen microbiota in Hu sheep

Objective: Effects of Lycium barbarum branches and leaves on slaughter performance, growth performance and apparent digestibility of nutritent, meat quality and sequencing of Hu sheep.

Point 1: Limitations: Check the various details mentioned in the comments and please also check why there is no explanation as to why the treatment with the lowest level of Lycium barbarum branches and leaves has an effect and the other concentrations show a lower effect on the studied variables.

Response 1: We checked the various details mentioned in the comments, and rewritten the summary results and conclusion sections to supplement the description of the different treatment groups.

Comments

Simple Summary: No comments

Abstract:

Point 2: Delete the word “Background”. What is the meaning of GR value?

Response 2: We have deleted the background. The GR value is an indicator of carcass fat content and has the same meaning as back fat thickness. The difference between the two is the location of the measurement.

Point 3: What is “positive effect on the growth, development? organ development” referring to?

Response 3: It means that adding appropriate amount of Lycium barbarum branches and leaves to the diet can promote the growth performance, feed efficiency and meat quality of Hu sheep. I have modified the description in the conclusion.

Point 4: Keywords: What is “slaughtering performance”? Does it refer to carcass quality?

Response 4: Slaughter performance, as a comprehensive indicator reflecting livestock production performance, has an important impact on the economic benefits of breeding. It is mainly reflected by indicators such as carcass weight, slaughter rate, bone-to-meat ratio, eye muscle area and backfat thickness. Slaughter rate and carcass weight, as its key indicators, can directly reflect the meat production capacity of livestock and poultry.

Introduction: No comments

Materials and Methods

Point 5: Lines 121-124: Is the process done twice?

Response 5: The digestion test was conducted once and consisted of ten days. The first five days were used to acclimatise the test sheep to the collection bag. On day 6, the collection of all the faeces of each sheep for one day was officially started to be analyzed for the subsequent test samples.

Point 6: Line 131: What does “GR value” mean?

Response 6: GR value is the total tissue thickness between the 12th and 13th ribs and 11 cm from the dorsal midline of the carcass. The GR value is an indicator of carcass fat content and has the same meaning as backfat thickness. The difference between the two is the location of the measurement.

Point 7: Line 142: Was metaphosphoric acid added for the measuement of volatile fatty acids?

Response 7: Metaphosphoric acid was added in the determination of ammonia nitrogen and volatile fatty acids in rumen fluid, and the specific sample processing information has been supplemented in the method.

Results

Point 8: Lines 211-226 and 234: The description of each variable obtained can be presented in the Materials and Methods section.

Response 8: We agree with the your suggestion and have rewritten this section accordingly. We have added specific explanations of the indicators of slaughter performance and meat quality and their determination, as well as a detailed description of the procedure for the treatment of rumen liquor.

Point 9: Table 5: The variable for the acetate/propionate ratio could be added.

Response 9: We have added a variable for the acetate/propionate ratio.

Discussion

Point 10: Line 351: The following statement “the F:G  ratio was significantly reduced” should be changed to “tends to” because the p-value was 0.068.

Response 10: Yes, you ' re right. I checked my data and changed the description to except for the control group, the F : G ratio tended to increase with the increase of the proportion of LBL.

Point 11: Please add the reference(s) for the following paragraph: “suggests that including LBL promotes the growth of Hu sheep, possibly due to the richness of LBL in polysaccharides, carotenoids, polyphenols, and flavonoids. These compounds enhance the fermentation of microorganisms in the rumen and improve the organism's ability to absorb nutrients, promoting the animal's growth performance.”

Response 11: We have added references.

Point 12: Line 366: Cite previous studies.

Response 12: We have added references.

Point 13: Line 370: What does LBP mean? Please add the full name.

Response 13: LBP refers to Lycium barbarum polysaccharide, which has been added in the full name of LBP.

Point 14: Lines 367-373: The following sentence is repeated three times. Please correct and add its reference(s). “It was discovered that LBP, an active ingredient found in LBL, can regulate intestinal flora, leading to an increase in the relative abundance of beneficial flora in the intestinal microorganisms.”

Response 14: Yes, we also found the error, deleted the duplicate statement, and added the reference.

Point 15: Lines 374-375: Add the corresponding citations.

Response 15: I'm sorry, but I don't know what the problem is. During the revision process, I realised that the line numbers don't match up. It seems to me that lines 374-375 are a sentence describing the results of this test and don't need a reference.

Point 16: Lines 342, 380, 383: The citations are made using the authors’ first names when it should be their last names. Please check and change all the corresponding citations in the whole document.

Response 16: We corrected this part.

Point 17: Line 392: In the following statement, “differences in the form”, does the word “form” refer to the type of active compound?

Response 17: The difference in form refers to the different forms added to the ration. Previous studies directly fed Lycium barbarum fruits or made extracts to be added, while the present experiment used Lycium barbarum branches and leaves as roughage to be fed directly.

Point 18: Lines 399-403: The arguments about LBP, even when they could be valid, are risky assumptions since there is no mention in the document on which concentrations of this compound were used in the treatment groups, neither theoretically nor practically.

Response 18:Thank you for your comments, we have modified this part.

Point 19: Lines 422-424: Even when propionate and valerate were increased during the experiment, it doesn’t mean that the total VFA concentration was increased as argued with the following statement “This result is consistent with the findings of Zhang et al. [52], who demonstrated that adding LBL to the diet could increase the total concentration of VFA in the rumen of Hu sheep”.

Response 19: Yes, you are right. I went and rechecked this reference and updated the description of the results in the text.

Point 20: Lines 427-430: The argument about the effect on the intestinal microorganisms has no basis or reference that supports it.

Response 20: We have added references on the effects on gut microbes.

Point 21: Lines 504-506: It is not the clear what the following sentence is supposed to expose “This suggests that an ingredient in the diets used in this study has the ability to promote the synthesis of antibiotics, although the specific mechanism of this effect is not yet known”.

Response 21: Thank you for your comments.We have deleted this part.

Conclusions

Point 22: What is meant with “considerable economic benefits”? What variable(s) does it refer to? Please consider to rewrite the Conclusions with a higher level of detail and specify the limitations of the study.

Response 22: This test did not measure the economic benefits, but only through the ratio of material to weight to infer. I also found that this statement is not accurate, the description in the conclusion has been modified, has been deleted ' considerable economic benefits '.

References

Point 23: Please check and homogenize the format: should the journal name be abbreviated? Use the same uppercase and lowercase format for it. Also check which part should be italicized and write the authors’ names starting with their last names, instead of their first names.

Response 23: Thank you for your comments. We checked all the references and modified the error part.

Round 2

Reviewer 2 Report

Comments and Suggestions for Authors

General comments.

The authors have done a good job of improving this paper particularly with the rewrite of the discussion. I was disappointed not to see you adopt a polynomial analysis to more clearly define the response to treatments, but the method you chose is an accepted statistical method. Specific comments.

Lines 125 to 132. You need to explain the meanings of the references to methods (e.g. GB / T  6438-2007 etc. I assume these are methods form a manual such as AOAC. Please cite the reference.

As previously, I cannot comment on the microbiology as I am not an expert in this field.

Author Response

Dear Editors and Reviewers:

Thank you for your comments concerning our manuscript entitled "Lycium barbarum branches and leaves enhance the growth performance and improve the rumen microbiota in Hu sheep”(ID: animals-2982614).Those comments are all valuable and very helpful for revising and improving our paper, as welll as the important guiding significance to our researches. We havestudied comments carefully and have made correction which we hope meet with approval. Revised portion are marked in red in the paper. Here you need to be explained that we have changed all the rumen bacteria in the article to italicised format and red. In addition, the main revisions of the paper and the responses to the reviewers' comments have been uploaded as an attachment.

Reviewer 3 Report

Comments and Suggestions for Authors

Title: Lycium barbarum branches and leaves enhance the growth performance and improve the rumen microbiota in Hu sheep

Objective: Effects of Lycium barbarum branches and leaves on slaughter performance, growth performance and apparent digestibility of nutrients, meat quality and sequencing of Hu sheep.

Limitations: Mention that more studies need to be carried out on the mechanism of action of Lycium barbarum branches and leaves in ruminants. As well as a greater number of studies at the molecular level of the expression of enzymatic activity of bacteria.

In line 8 the initials of the author Lihong Hu should be (L.H) and not (H.L)

Line 38, the abbreviation of GR include its full description before the abbreviation in the text of the summary and in the document, carcass fat content.

Lines 38-39, There was no significant difference in organ weight and organ index between the experimental groups (p < 0.05). Then it should be changed to (p > 0.05)

Lines 41-42, There was no significant difference in NH3-N content between the experimental groups (p < 0.05). Then it should be changed to (p > 0.05).

Lines 44-45, In addition, this had no significant effect on the structure and abundance of rumen microbiota (p < 0.05). Then it should be changed to (p > 0.05).

Lines 370-372, Additionally, except for the control group, the F: G ratio tended to decrease with the increase of the proportion of LBL, and the F: G ratio of LBL2 group was the lowest. Check the text, as only LBL1 and LBL2 showed the effect. Furthermore, the LBL1 treatment was the lowest.

Line 395 and 413, VFA change to VFAs

Line 500, 4 k/kg LBP, what is k? Would it be g/kg?

Please check and homogenize the format: should the journal name be abbreviated? If applicable, check that the name of the journal is not abbreviated in some references.

Author Response

(The authors gave the same response as above.)
